# Exploring Models and Data for Image Question Answering

**Mengye Ren**[1], **Ryan Kiros**[1], **Richard S. Zemel**[1,2]
University of Toronto[1]
Canadian Institute for Advanced Research[2]
{mren, rkiros, zemel}@cs.toronto.edu

## Abstract

This work aims to address the problem of image-based question-answering (QA) with new models and datasets. In our work, we propose to use neural networks and visual semantic embeddings, without intermediate stages such as object detection and image segmentation, to predict answers to simple questions about images. Our model performs 1.8 times better than the only published results on an existing image QA dataset. We also present a question generation algorithm that converts image descriptions, which are widely available, into QA form. We used this algorithm to produce an order-of-magnitude larger dataset, with more evenly distributed answers. A suite of baseline results on this new dataset are also presented.

## 1 Introduction

Combining image understanding and natural language interaction is one of the grand dreams of artificial intelligence. We are interested in the problem of jointly learning image and text through a question-answering task. Recently, researchers studying image caption generation [1, 2, 3, 4, 5, 6, 7, 8, 9, 10] have developed powerful methods of jointly learning from image and text inputs to form higher level representations from models such as convolutional neural networks (CNNs) trained on object recognition, and word embeddings trained on large scale text corpora. Image QA involves an extra layer of interaction between human and computers. Here the model needs to pay attention to details of the image instead of describing it in a vague sense. The problem also combines many computer vision sub-problems such as image labeling and object detection.

In this paper we present our contributions to the problem: a generic end-to-end QA model using visual semantic embeddings to connect a CNN and a recurrent neural net (RNN), as well as comparisons to a suite of other models; an automatic question generation algorithm that converts description sentences into questions; and a new QA dataset (COCO-QA) that was generated using the algorithm, and a number of baseline results on this new dataset.

In this work we assume that the answers consist of only a single word, which allows us to treat the problem as a classification problem. This also makes the evaluation of the models easier and more robust, avoiding the thorny evaluation issues that plague multi-word generation problems.

## 2 Related Work

Malinowski and Fritz [11] released a dataset with images and question-answer pairs, the DAtaset for QUestion Answering on Real-world images (DAQUAR). All images are from the NYU depth v2 dataset [12], and are taken from indoor scenes. Human segmentation, image depth values, and object labeling are available in the dataset. The QA data has two sets of configurations, which differ by the

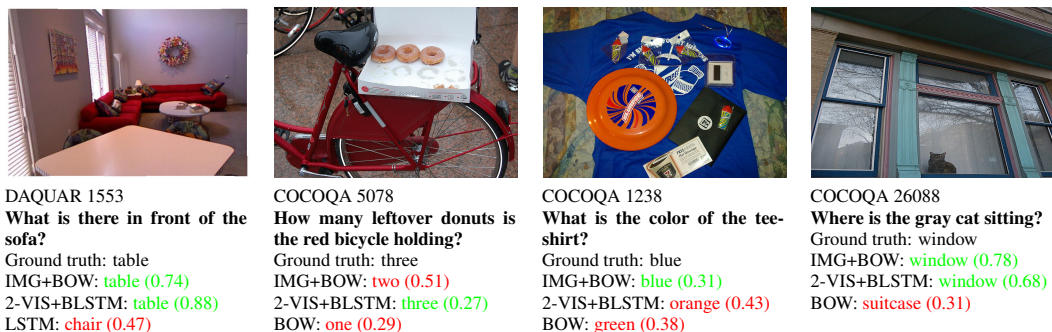

DAQUAR 1553
**What is there in front of the sofa?**
Ground truth: table
IMG+BOW: table (0.74)
2-VIS+BLSTM: table (0.88)
LSTM: chair (0.47)

COCOQA 5078
**How many leftover donuts is the red bicycle holding?**
Ground truth: three
IMG+BOW: two (0.51)
2-VIS+BLSTM: three (0.27)
BOW: one (0.29)

COCOQA 1238
**What is the color of the tee-shirt?**
Ground truth: blue
IMG+BOW: blue (0.31)
2-VIS+BLSTM: orange (0.43)
BOW: green (0.38)

COCOQA 26088
**Where is the gray cat sitting?**
Ground truth: window
IMG+BOW: window (0.78)
2-VIS+BLSTM: window (0.68)
BOW: suitcase (0.31)

Figure 1: Sample questions and responses of a variety of models. Correct answers are in green and incorrect in red. The numbers in parentheses are the probabilities assigned to the top-ranked answer by the given model. The leftmost example is from the DAQUAR dataset, and the others are from our new COCO-QA dataset.

number of object classes appearing in the questions (37-class and 894-class). There are mainly three types of questions in this dataset: object type, object color, and number of objects. Some questions are easy but many questions are very hard to answer even for humans. Since DAQUAR is the only publicly available image-based QA dataset, it is one of our benchmarks to evaluate our models.

Together with the release of the DAQUAR dataset, Malinowski and Fritz presented an approach which combines semantic parsing and image segmentation. Their approach is notable as one of the first attempts at image QA, but it has a number of limitations. First, a human-defined possible set of predicates are very dataset-specific. To obtain the predicates, their algorithm also depends on the accuracy of the image segmentation algorithm and image depth information. Second, their model needs to compute all possible spatial relations in the training images. Even though the model limits this to the nearest neighbors of the test images, it could still be an expensive operation in larger datasets. Lastly the accuracy of their model is not very strong. We show below that some simple baselines perform better.

Very recently there has been a number of parallel efforts on both creating datasets and proposing new models [13, 14, 15, 16]. Both Antol et al. [13] and Gao et al. [15] used MS-COCO [17] images and created an open domain dataset with human generated questions and answers. In Anto et al.'s work, the authors also included cartoon pictures besides real images. Some questions require logical reasoning in order to answer correctly. Both Malinowski et al. [14] and Gao et al. [15] use recurrent networks to encode the sentence and output the answer. Whereas Malinowski et al. use a single network to handle both encoding and decoding, Gao et al. used two networks, a separate encoder and decoder. Lastly, bilingual (Chinese and English) versions of the QA dataset are available in Gao et al.'s work. Ma et al. [16] use CNNs to both extract image features and sentence features, and fuse the features together with another multi-modal CNN.

Our approach is developed independently from the work above. Similar to the work of Malinowski et al. and Gao et al., we also experimented with recurrent networks to consume the sequential question input. Unlike Gao et al., we formulate the task as a classification problem, as there is no single well- accepted metric to evaluate sentence-form answer accuracy [18]. Thus, we place more focus on a limited domain of questions that can be answered with one word. We also formulate and evaluate a range of other algorithms, that utilize various representations drawn from the question and image, on these datasets.

## 3 Proposed Methodology

The methodology presented here is two-fold. On the model side we develop and apply various forms of neural networks and visual-semantic embeddings on this task, and on the dataset side we propose new ways of synthesizing QA pairs from currently available image description datasets.

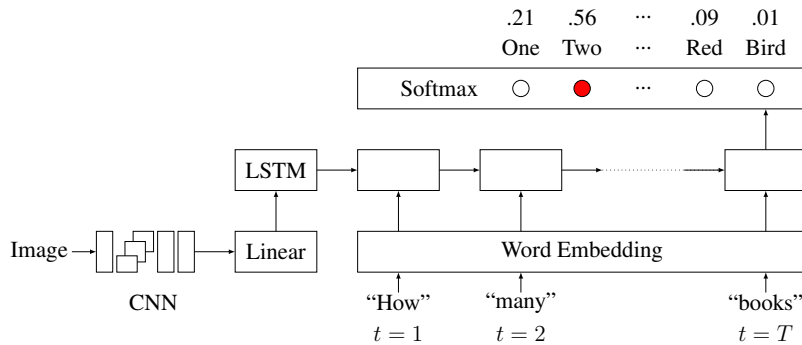

Figure 2: VIS+LSTM Model

## 3.1 Models

In recent years, recurrent neural networks (RNNs) have enjoyed some successes in the field of natural language processing (NLP). Long short-term memory (LSTM) [19] is a form of RNN which is easier to train than standard RNNs because of its linear error propagation and multiplicative gatings. Our model builds directly on top of the LSTM sentence model and is called the "VIS+LSTM" model. It treats the image as one word of the question. We borrowed this idea of treating the image as a word from caption generation work done by Vinyals et al. [1]. We compare this newly proposed model with a suite of simpler models in the Experimental Results section.

1. We use the last hidden layer of the 19-layer Oxford VGG Conv Net [20] trained on ImageNet 2014 Challenge [21] as our visual embeddings. The CNN part of our model is kept frozen during training.

2. We experimented with several different word embedding models: randomly initialized embedding, dataset-specific skip-gram embedding and general-purpose skip-gram embedding model [22]. The word embeddings are trained with the rest of the model.

3. We then treat the image as if it is the first word of the sentence. Similar to DeViSE [23], we use a linear or affine transformation to map 4096 dimension image feature vectors to a 300 or 500 dimensional vector that matches the dimension of the word embeddings.

4. We can optionally treat the image as the last word of the question as well through a different weight matrix and optionally add a reverse LSTM, which gets the same content but operates in a backward sequential fashion.

5. The LSTM(s) outputs are fed into a softmax layer at the last timestep to generate answers.

## 3.2 Question-Answer Generation

The currently available DAQUAR dataset contains approximately 1500 images and 7000 questions on 37 common object classes, which might be not enough for training large complex models. Another problem with the current dataset is that simply guessing the modes can yield very good accuracy.

We aim to create another dataset, to produce a much larger number of QA pairs and a more even distribution of answers. While collecting human generated QA pairs is one possible approach, and another is to synthesize questions based on image labeling, we instead propose to automatically convert descriptions into QA form. In general, objects mentioned in image descriptions are easier to detect than the ones in DAQUAR's human generated questions, and than the ones in synthetic QAs based on ground truth labeling. This allows the model to rely more on rough image understanding without any logical reasoning. Lastly the conversion process preserves the language variability in the original description, and results in more human-like questions than questions generated from image labeling.

As a starting point we used the MS-COCO dataset [17], but the same method can be applied to any other image description dataset, such as Flickr [24], SBU [25], or even the internet.

### 3.2.1 Pre-Processing & Common Strategies

We used the Stanford parser [26] to obtain the syntatic structure of the original image description. We also utilized these strategies for forming the questions.

1. Compound sentences to simple sentences
   Here we only consider a simple case, where two sentences are joined together with a conjunctive word. We split the orginial sentences into two independent sentences.
2. Indefinite determiners "a(n)" to definite determiners "the".
3. Wh-movement constraints
   In English, questions tend to start with interrogative words such as "what". The algorithm needs to move the verb as well as the "wh-" constituent to the front of the sentence. For example: "A man is riding a **horse**" becomes "**What** is **the** man riding?" In this work we consider the following two simple constraints: (1) A-over-A principle which restricts the movement of a wh-word inside a noun phrase (NP) [27]; (2) Our algorithm does not move any wh-word that is contained in a clause constituent.

### 3.2.2 Question Generation

Question generation is still an open-ended topic. Overall, we adopt a conservative approach to generating questions in an attempt to create high-quality questions. We consider generating four types of questions below:

1. *Object* Questions: First, we consider asking about an object using "what". This involves replacing the actual object with a "what" in the sentence, and then transforming the sentence structure so that the "what" appears in the front of the sentence. The entire algorithm has the following stages: (1) Split long sentences into simple sentences; (2) Change indefinite determiners to definite determiners; (3) Traverse the sentence and identify potential answers and replace with "what". During the traversal of object-type question generation, we currently ignore all the prepositional phrase (PP) constituents; (4) Perform wh-movement. In order to identify a possible answer word, we used WordNet [28] and the NLTK software package [29] to get noun categories.
2. *Number* Questions: We follow a similar procedure as the previous algorithm, except for a different way to identify potential answers: we extract numbers from original sentences. Splitting compound sentences, changing determiners, and wh-movement parts remain the same.
3. *Color* Questions: Color questions are much easier to generate. This only requires locating the color adjective and the noun to which the adjective attaches. Then it simply forms a sentence "What is the color of the [object]" with the "object" replaced by the actual noun.
4. *Location* Questions: These are similar to generating object questions, except that now the answer traversal will only search within PP constituents that start with the preposition "in". We also added rules to filter out clothing so that the answers will mostly be places, scenes, or large objects that contain smaller objects.

### 3.2.3 Post-Processing

We rejected the answers that appear too rarely or too often in our generated dataset. After this QA rejection process, the frequency of the most common answer words was reduced from 24.98% down to 7.30% in the test set of COCO-QA.

## 4 Experimental Results

### 4.1 Datasets

Table 1 summarizes the statistics of COCO-QA. It should be noted that since we applied the QA pair rejection process, mode-guessing performs very poorly on COCO-QA. However, COCO-QA questions are actually easier to answer than DAQUAR from a human point of view. This encourages the model to exploit salient object relations instead of exhaustively searching all possible relations. COCO-QA dataset can be downloaded at `http://www.cs.toronto.edu/~mren/imageqa/data/cocoqa`

Table 1: COCO-QA question type break-down

| CATEGORY | TRAIN | % | TEST | % |
|---|---|---|---|---|
| OBJECT | 54992 | 69.84% | 27206 | 69.85% |
| NUMBER | 5885 | 7.47% | 2755 | 7.07% |
| COLOR | 13059 | 16.59% | 6509 | 16.71% |
| LOCATION | 4800 | 6.10% | 2478 | 6.36% |
| TOTAL | 78736 | 100.00% | 38948 | 100.00% |

Here we provide some brief statistics of the new dataset. The maximum question length is 55, and average is 9.65. The most common answers are "two" (3116, 2.65%), "white" (2851, 2.42%), and "red" (2443, 2.08%). The least common are "eagle" (25, 0.02%) "tram" (25, 0.02%), and "sofa" (25, 0.02%). The median answer is "bed" (867, 0.737%). Across the entire test set (38,948 QAs), 9072 (23.29%) overlap in training questions, and 7284 (18.70%) overlap in training question-answer pairs.

### 4.2 Model Details

1. **VIS+LSTM**: The first model is the CNN and LSTM with a dimensionality-reduction weight matrix in the middle; we call this "VIS+LSTM" in our tables and figures.

2. **2-VIS+BLSTM**: The second model has two image feature inputs, at the start and the end of the sentence, with different learned linear transformations, and also has LSTMs going in both the forward and backward directions. Both LSTMs output to the softmax layer at the last timestep. We call the second model "2-VIS+BLSTM".

3. **IMG+BOW**: This simple model performs multinomial logistic regression based on the image features without dimensionality reduction (4096 dimension), and a bag-of-word (BOW) vector obtained by summing all the learned word vectors of the question.

4. **FULL**: Lastly, the "FULL" model is a simple average of the three models above.

We release the complete details of the models at `https://github.com/renmengye/imageqa-public`.

### 4.3 Baselines

To evaluate the effectiveness of our models, we designed a few baselines.

1. **GUESS**: One very simple baseline is to predict the mode based on the question type. For example, if the question contains "how many" then the model will output "two." In DAQUAR, the modes are "table", "two", and "white" and in COCO-QA, the modes are "cat", "two", "white", and "room".

2. **BOW**: We designed a set of "blind" models which are given only the questions without the images. One of the simplest blind models performs logistic regression on the BOW vector to classify answers.

3. **LSTM**: Another "blind" model we experimented with simply inputs the question words into the LSTM alone.

4. **IMG**: We also trained a counterpart "deaf" model. For each type of question, we train a separate CNN classification layer (with all lower layers frozen during training). Note that this model knows the type of question, in order to make its performance somewhat comparable to models that can take into account the words to narrow down the answer space. However the model does not know anything about the question except the type.

5. **IMG+PRIOR**: This baseline combines the prior knowledge of an object and the image understanding from the "deaf model". For example, a question asking the color of a white bird flying in the blue sky may output white rather than blue simply because the prior probability of the bird being blue is lower. We denote $c$ as the color, $o$ as the class of the object of interest, and $x$ as the

image. Assuming $o$ and $x$ are conditionally independent given the color,

$$p(c|o, x) = \frac{p(c, o|x)}{\sum_{c \in \mathcal{C}} p(c, o|x)} = \frac{p(o|c, x)p(c|x)}{\sum_{c \in \mathcal{C}} p(o|c, x)p(c|x)} = \frac{p(o|c)p(c|x)}{\sum_{c \in \mathcal{C}} p(o|c)p(c|x)} \quad (1)$$

This can be computed if $p(c|x)$ is the output of a logistic regression given the CNN features alone, and we simply estimate $p(o|c)$ empirically: $\hat{p}(o|c) = \frac{count(o,c)}{count(c)}$. We use Laplace smoothing on this empirical distribution.

6. **K-NN**: In the task of image caption generation, Devlin et al. [30] showed that a nearest neighbors baseline approach actually performs very well. To see whether our model memorizes the training data for answering new question, we include a K-NN baseline in the results. Unlike image caption generation, here the similarity measure includes both image and text. We use the bag-of-words representation learned from IMG+BOW, and append it to the CNN image features. We use Euclidean distance as the similarity metric; it is possible to improve the nearest neighbor result by learning a similarity metric.

## 4.4 Performance Metrics

To evaluate model performance, we used the plain answer accuracy as well as the Wu-Palmer similarity (WUPS) measure [31, 32]. The WUPS calculates the similarity between two words based on their longest common subsequence in the taxonomy tree. If the similarity between two words is less than a threshold then a score of zero will be given to the candidate answer. Following Malinowski and Fritz [32], we measure all models in terms of accuracy, WUPS 0.9, and WUPS 0.0.

## 4.5 Results and Analysis

Table 2 summarizes the learning results on DAQUAR and COCO-QA. For DAQUAR we compare our results with [32] and [14]. It should be noted that our DAQUAR results are for the portion of the dataset (98.3%) with single-word answers. After the release of our paper, Ma et al. [16] claimed to achieve better results on both datasets.

Table 2: DAQUAR and COCO-QA results

| | DAQUAR | | | COCO-QA | | |
|---|---|---|---|---|---|---|
| | ACC. | WUPS 0.9 | WUPS 0.0 | ACC. | WUPS 0.9 | WUPS 0.0 |
| MULTI-WORLD [32] | 0.1273 | 0.1810 | 0.5147 | - | - | - |
| GUESS | 0.1824 | 0.2965 | 0.7759 | 0.0730 | 0.1837 | 0.7413 |
| BOW | 0.3267 | 0.4319 | 0.8130 | 0.3752 | 0.4854 | 0.8278 |
| LSTM | 0.3273 | 0.4350 | 0.8162 | 0.3676 | 0.4758 | 0.8234 |
| IMG | - | - | - | 0.4302 | 0.5864 | 0.8585 |
| IMG+PRIOR | - | - | - | 0.4466 | 0.6020 | 0.8624 |
| K-NN (K=31, 13) | 0.3185 | 0.4242 | 0.8063 | 0.4496 | 0.5698 | 0.8557 |
| IMG+BOW | 0.3417 | 0.4499 | 0.8148 | **0.5592** | **0.6678** | **0.8899** |
| VIS+LSTM | 0.3441 | 0.4605 | **0.8223** | 0.5331 | 0.6391 | 0.8825 |
| ASK-NEURON [14] | 0.3468 | 0.4076 | 0.7954 | - | - | - |
| 2-VIS+BLSTM | **0.3578** | **0.4683** | 0.8215 | 0.5509 | 0.6534 | 0.8864 |
| FULL | **0.3694** | **0.4815** | **0.8268** | **0.5784** | **0.6790** | **0.8952** |
| HUMAN | 0.6027 | 0.6104 | 0.7896 | - | - | - |

From the above results we observe that our model outperforms the baselines and the existing approach in terms of answer accuracy and WUPS. Our VIS+LSTM and Malinkowski et al.'s recurrent neural network model [14] achieved somewhat similar performance on DAQUAR. A simple average of all three models further boosts the performance by 1-2%, outperforming other models.

It is surprising to see that the IMG+BOW model is very strong on both datasets. One limitation of our VIS+LSTM model is that we are not able to consume image features as large as 4096 dimensions at one time step, so the dimensionality reduction may lose some useful information. We tried to give IMG+BOW a 500 dim. image vector, and it does worse than VIS+LSTM ($\approx$48%).

Table 3: COCO-QA accuracy per category

|  | OBJECT | NUMBER | COLOR | LOCATION |
|---|---|---|---|---|
| GUESS | 0.0239 | 0.3606 | 0.1457 | 0.0908 |
| BOW | 0.3727 | 0.4356 | 0.3475 | 0.4084 |
| LSTM | 0.3587 | 0.4534 | 0.3626 | 0.3842 |
| IMG | 0.4073 | 0.2926 | 0.4268 | 0.4419 |
| IMG+PRIOR | - | 0.3739 | 0.4899 | 0.4451 |
| K-NN | 0.4799 | 0.3699 | 0.3723 | 0.4080 |
| IMG+BOW | **0.5866** | 0.4410 | **0.5196** | **0.4939** |
| VIS+LSTM | 0.5653 | **0.4610** | 0.4587 | 0.4552 |
| 2-VIS+BLSTM | 0.5817 | 0.4479 | 0.4953 | 0.4734 |
| FULL | **0.6108** | **0.4766** | 0.5148 | **0.5028** |

By comparing the blind versions of the BOW and LSTM models, we hypothesize that in Image QA tasks, and in particular on the simple questions studied here, sequential word interaction may not be as important as in other natural language tasks.

It is also interesting that the blind model does not lose much on the DAQUAR dataset, We speculate that it is likely that the ImageNet images are very different from the indoor scene images, which are mostly composed of furniture. However, the non-blind models outperform the blind models by a large margin on COCO-QA. There are three possible reasons: (1) the objects in MS-COCO resemble the ones in ImageNet more; (2) MS-COCO images have fewer objects whereas the indoor scenes have considerable clutter; and (3) COCO-QA has more data to train complex models.

There are many interesting examples but due to space limitations we can only show a few in Figure 1 and Figure 3; full results are available at `http://www.cs.toronto.edu/~mren/ imageqa/results`. For some of the images, we added some extra questions (the ones have an "a" in the question ID); these provide more insight into a model's representation of the image and question information, and help elucidate questions that our models may accidentally get correct. The parentheses in the figures represent the confidence score given by the softmax layer of the respective model.

**Model Selection**: We did not find that using different word embedding has a significant impact on the final classification results. We observed that fine-tuning the word embedding results in better performance and normalizing the CNN hidden image features into zero-mean and unit-variance helps achieve faster training time. The bidirectional LSTM model can further boost the result by a little.

**Object Questions**: As the original CNN was trained for the ImageNet challenge, the IMG+BOW benefited significantly from its single object recognition ability. However, the challenging part is to consider spatial relations between multiple objects and to focus on details of the image. Our models only did a moderately acceptable job on this; see for instance the first picture of Figure 1 and the fourth picture of Figure 3. Sometimes a model fails to make a correct decision but outputs the most salient object, while sometimes the blind model can equally guess the most probable objects based on the question alone (e.g., chairs should be around the dining table). Nonetheless, the FULL model improves accuracy by 50% compared to IMG model, which shows the difference between pure object classification and image question answering.

**Counting**: In DAQUAR, we could not observe any advantage in the counting ability of the IMG+BOW and the VIS+LSTM model compared to the blind baselines. In COCO-QA there is some observable counting ability in very clean images with a single object type. The models can sometimes count up to five or six. However, as shown in the second picture of Figure 3, the ability is fairly weak as they do not count correctly when different object types are present. There is a lot of room for improvement in the counting task, and in fact this could be a separate computer vision problem on its own.

**Color**: In COCO-QA there is a significant win for the IMG+BOW and the VIS+LSTM against the blind ones on color-type questions. We further discovered that these models are not only able to recognize the dominant color of the image but sometimes associate different colors to different objects, as shown in the first picture of Figure 3. However, they still fail on a number of easy

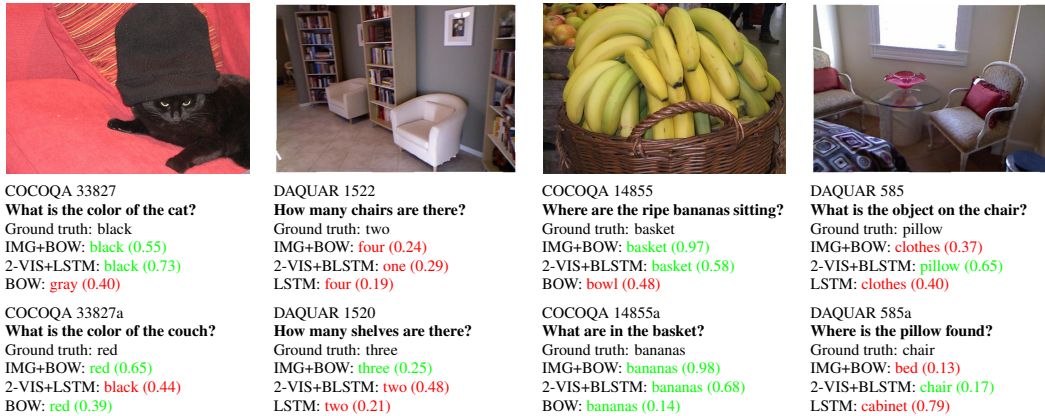

COCOQA 33827
**What is the color of the cat?**
Ground truth: black
IMG+BOW: black (0.55)
2-VIS+LSTM: black (0.73)
BOW: gray (0.40)

COCOQA 33827a
**What is the color of the couch?**
Ground truth: red
IMG+BOW: red (0.65)
2-VIS+LSTM: black (0.44)
BOW: red (0.39)

DAQUAR 1522
**How many chairs are there?**
Ground truth: two
IMG+BOW: four (0.24)
2-VIS+BLSTM: one (0.29)
LSTM: four (0.19)

DAQUAR 1520
**How many shelves are there?**
Ground truth: three
IMG+BOW: three (0.25)
2-VIS+BLSTM: two (0.48)
LSTM: two (0.21)

COCOQA 14855
**Where are the ripe bananas sitting?**
Ground truth: basket
IMG+BOW: basket (0.97)
2-VIS+BLSTM: basket (0.58)
BOW: bowl (0.48)

COCOQA 14855a
**What are in the basket?**
Ground truth: bananas
IMG+BOW: bananas (0.98)
2-VIS+BLSTM: bananas (0.68)
BOW: bananas (0.14)

DAQUAR 585
**What is the object on the chair?**
Ground truth: pillow
IMG+BOW: clothes (0.37)
2-VIS+BLSTM: pillow (0.65)
LSTM: clothes (0.40)

DAQUAR 585a
**Where is the pillow found?**
Ground truth: chair
IMG+BOW: bed (0.13)
2-VIS+BLSTM: chair (0.17)
LSTM: cabinet (0.79)

Figure 3: Sample questions and responses of our system

examples. Adding prior knowledge provides an immediate gain on the IMG model in terms of accuracy on Color and Number questions. The gap between the IMG+PRIOR and IMG+BOW shows some localized color association ability in the CNN image representation.

# 5 Conclusion and Current Directions

In this paper, we consider the image QA problem and present our end-to-end neural network models. Our model shows a reasonable understanding of the question and some coarse image understanding, but it is still very naïve in many situations. While recurrent networks are becoming a popular choice for learning image and text, we showed that a simple bag-of-words can perform equally well compared to a recurrent network that is borrowed from an image caption generation framework [1]. We proposed a more complete set of baselines which can provide potential insight for developing more sophisticated end-to-end image question answering systems. As the currently available dataset is not large enough, we developed an algorithm that helps us collect large scale image QA dataset from image descriptions. Our question generation algorithm is extensible to many image description datasets and can be automated without requiring extensive human effort. We hope that the release of the new dataset will encourage more data-driven approaches to this problem in the future.

Image question answering is a fairly new research topic, and the approach we present here has a number of limitations. First, our models are just answer classifiers. Ideally we would like to permit longer answers which will involve some sophisticated text generation model or structured output. But this will require an automatic free-form answer evaluation metric. Second, we are only focusing on a limited domain of questions. However, this limited range of questions allow us to study the results more in depth. Lastly, it is also hard to interpret why the models output a certain answer. By comparing our models with some baselines we can roughly infer whether they understood the image. Visual attention is another future direction, which could both improve the results (based on recent successes in image captioning [8]) as well as help explain the model prediction by examining the attention output at every timestep.

### Acknowledgments

We would like to thank Nitish Srivastava for the support of Toronto Conv Net, from which we extracted the CNN image features. We would also like to thank anonymous reviewers for their valuable and helpful comments.

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
