[Reviews · NeurIPS 2015]

Submitted by Assigned_Reviewer_1

Great paper introducing a new dataset for vision + NLP.

Quality: Good.

Clarity: The paper is well written, though the discussion in L349 - L370 should reference specific instances in the figure. Currently the text just states, e.g., "Fig 3 shows..." without pointing out which of the 24 numbers in Fig 3 we should be comparing or contrasting to each other.

Originality: The models are not original, but the dataset is. The originality is defended by comparison to DAQUAR. The paper claims that there are only minor difference between blind vs. non-blind models when tested on DAQUAR which motivates this new dataset which requires more thorough visual attention to attain high test accuracy.

Significance: Likely to be received positively by the field and promote further work merging vision + NLP models.

As one last comment, I found that the Number questions often had grammatically incorrect English:

"What does the man rid while wearing..." (wrong verb) "How many double deckered busses parked near the green truck?" (missing verb)

On the other hand, Color and Location questions seemed to be grammatically correct. Perhaps the Number questions should be read and vetted or corrected by mechanical Turkers.
Summary: Great paper introducing a new dataset for vision + NLP.

Submitted by Assigned_Reviewer_2

Summary:

This paper addresses the task of image-based Q&A on 2 axes: comparison of different models on 2 datasets and creation of a new dataset based on existing captions.

Quality:

The paper is addressing an important and interesting new topic which has seen recent surge of interest (Malinowski2014, Malinowski2015, Antol2015, Gao2015, etc.). The paper is technically sound, well-written, and well-organized. They achieve good results on both datasets and the baselines are useful to understand important ablations. The new dataset is also much larger than previous work, allowing training of stronger models, esp. deep NN ones.

However, there are several weaknesses: their main model is not very different from existing work on image-Q&A (Malinowski2015, who also had a VIS+LSTM style model (but they were also jointly training the CNN and RNN, and also decoding with RNNs to produce longer answers) and achieves similar performance (except that adding bidirectionality and 2-way image input helps). Also, as the authors themselves discuss, the dataset in its current form, synthetically created from captions, is a good start but is quite conservative and limited, being single-word answers, and the transformation rules only designed for certain simple syntactic cases.

It is exploration work and will benefit a lot from a bit more progress in terms of new models and a slightly more broad dataset (at least with answers up to 2-3 words).

Regarding new models, e.g., attention-based models are very relevant and intuitive here (and the paper would be much more complete with this), since these models should learn to focus on the right area of the image to answer the given question and it would be very interesting to analyze the results of whether this focusing happens correctly.

Before attention models, since 2-way image input helped (actually, it would be good to ablate 2-way versus bidirectionality in the 2-VIS+BLSTM model), it would be good to also show the model version that feeds the image vector at every time step of the question.

Also, it would be useful to have a nearest neighbor baseline as in Devlin et al., 2015, given their discussion of COCO's properties. Here too, one could imagine copying answers of training questions, for cases where the captions are very similar.

Regarding a broader-scope dataset, the issue with the current approach is that it is too similar to the captioning approach or task, which has the drawback that a major motivation to move to image-Q&A is to move away from single, vague (non-specific), generic, one-event-focused captions to a more complex and detailed understanding of and reasoning over the image; which doesn't happen with this paper's current dataset creation approach, and so this will also not encourage thinking of very different models to handle image-Q&A, since the best captioning models will continue to work well here. Also, having 2-3 word answers will capture more realistic and more diverse scenarios; and though it is true that evaluation is harder, one can start with existing metrics like BLEU, METEOR, CIDEr, and human eval. And since these will not be full sentences but just 2-3 word phrases, such existing metrics will be much more robust and stable already.

Some other comments:

-- Sec 4.2.1: The authors say splitting happens on conjunctions to give 'two independent sentences', but what if these creates a sentence and a clause, instead of two sentences? -- Sec 4.2.1: Indefinite determiners "a(n)" to definite determiners "the": what if the caption had something plural without an 'a', e.g., "two boys are playing in the park" and the question still needs a 'the', i.e., "where are the two boys playing?" -- Sec 5.5: IMG+BOW works well and the authors discuss the reasons for this to be the noisy dim-reduction of 4096-dim image vectors, and that the sequential RNNs are not needed for Q&A. However, this needs to ablated better to know which of these 2 reasons is more true, e.g., by having a IMG+BOW baseline on 300-500-dim image vectors instead of 4096-dim, and/or by using 4096-dim word embeddings (which should most likely be similarly accurate to 500-dim ones) in RNNs with the 4096-dim image vector.

References: Exploring Nearest Neighbor Approaches for Image Captioning Jacob Devlin, Saurabh Gupta, Ross Girshick, Margaret Mitchell, C. Lawrence Zitnick

Clarity:

The paper is well-written and well-organized.

Originality:

The task of image-Q&A is very recent with only a couple of prior and concurrent work, and the dataset creation procedure, despite its limitations (discussed above) is novel. The models are mostly not novel, being very similar to Malinowski2015, but the authors add bidirectionality and 2-way image input (but then Malinowski2015 was jointly training the CNN and RNN, and also decoding with RNNs to produce longer answers).

Significance:

As discussed above, the paper show useful results and ablations on the important, recent task of image-Q&A, based on 2 datasets -- an existing small dataset and a new large dataset; however, the second, new dataset is synthetically created by rule-transforming captions and only to single-word answers, thus keeping the impact of the dataset limited, because it keeps the task too similar to the generic captioning task and because there is no generation of answers or prediction of multi-word answers.
Summary: This paper addresses the important and interesting recent topic of image-Q&A, presenting multiple models and a new, larger, dataset. The paper is well-written and achieves good results on both datasets. However, the dataset in its current form is quite

limited because it just rule-converts captions to questions and only to single-word answers, hence keeping the task too similar to the caption generation task (issues in which were a major motivation to move to image-Q&A in the first place). Also, their main model is not very different from existing work on image-Q&A (Malinowski2015).

Submitted by Assigned_Reviewer_3

This paper addresses the problem of image question answering by proposing a neural network architecture based on an LSTM that uses pre-trained CNN based image features and word embeddings to produce answers to single word questions. The paper also provides some specific methods to generate questions through converting image descriptions into questions. The paper emphasizes how the reformulation of image annotation into Image QA allows for evaluation metrics that are easier to interpret compared to those used in the setting of multi-word image description. The proposed methods for question generation allow for the creation questions pertaining to: objects, numbers, colors and locations.

This point about formulating the problem of deeper visual understanding as a question answering problem for which evaluation metrics are much easier to work with is quite compelling in my view. The paper makes the interesting observation that the DAQUAR dataset is the only publically available image-based QA dataset - a situation that is addressed by the work here. The fact that his paper provides both a clear methodology of synthesizing QA pairs from currently available image description datasets, a straightforward novel neural architecture & appropriate baseline comparisons makes this paper quite useful. The question transformation methodology proposed here is itself potentially quite useful in my view.

However, in light of the potential utility of the aspect of the contribution here pertaining to question generation, it would be nice if this paper could just place the proposed strategy in context with some of the prior work on question generation.

While the underlying framework for dealing with the image as a word is borrowed from the caption generation work in [1], the general approach for the question answering model outlined in figure 2 is novel and quite interesting.

The inclusion of the IMG+BOW experiments provides a very helpful point of comparison & as noted by in the paper, it does surprisingly well for these questions. I am very happy that this baseline was given. However, given that the IMG+BOW appears to perform quite well and it is possibly within the range of uncertainty if such an analysis were to be provided, could the authors comment on the issue of significance testing with these results, ex. what sort of difference is needed to be significant on DAQUAR vs COCO-QA?

Summary: This paper provides a method to re-formulate image annotation datasets as question and answering datasets. A novel model for image QA is also given. The paper is clear, the proposed method and methodology is novel, the work provides considerable value to the community and it therefore should be accepted.

Submitted by Assigned_Reviewer_4

1. Summary of paper The paper addresses the problem of image question answering. The main contributions of the paper are: -- A neural network (CNN embedding + LSTM) based end to end approach for classifying answers to a given question (object, color, counting, location type questions only) given the corresponding image. The dataset used for testing the approach is DAQUAR dataset and COCO-QA dataset (a contribution of this paper). The results achieved are 1.8 times better than the only published results on DAQUAR dataset and slightly better than a parallel work on DAQUAR dataset. -- An algorithm to automatically convert image captions into questions of types- object, color, counting and location. -- A new COCO-QA dataset generated by using the above algorithm on MSCOCO captions dataset. -- Some baselines to compare the approach presented in the paper for image question answering.

2. Quality The task of answering natural language questions about images is a fairly new and active field of research in computer vision community. Recently there has been some parallel efforts towards both creating an image question answering dataset and exploring models to solve the problem. This paper compares the approach presented with a published work and also with a recent effort but unpublished work.

In this paper, the task of question answering has been treated as a classification problem as opposed to answer generation problem. Also, the presented approach tackles only a few types of questions namely object, color, counting and location based questions. But the authors are aware of these limitations of the paper. The paper evaluates several baseline methods to dig deeper into what information is helping the model to classify the answers. These baselines prove to be helpful. However, it would also be interesting to see SVM based baseline for IMG + BOW.

The COCO-QA dataset introduced in the paper is derived from the MSCOCO captions dataset which tend to be generic. So there will be a lot of similar questions in the dataset (since there are only 4 types of questions). If there are a large number of images about a particular object (e.g. giraffe) in the MSCOCO caption dataset, there will be a high proportion of question in the COCO-QA dataset of the type "what are eating leaves from trees" and the answer would be "giraffes". Although the authors control the number the questions in the dataset that have same answers, I still think that such questions will have less variety unlike human generated questions which will tend to be more specific (even for a particular question type such as object).

3. Clarity The paper lacks clarity about a number of technical details. Following are the details missing from the paper: -- What are the different classes of answers for the classification task? How is the evaluation done when a generated answer does not belong to any of the classes but is a correct answer (matches with ground truth)? -- How is the LSTM trained, what hyper parameters are used? -- Lines 204-205, what do the authors mean by "to identify a possible answer word"? -- Line 214, why is only "in" preposition used to generate location based questions? -- Section 4.2.3 (Line 229), is the percentage reduction in the mode component within each category of questions presented in Table 1 or across all questions. -- FULL model: What is this model exactly? What does the average of models mean?

-- Line 276- I don't buy the assumption that object and image are conditionally independent given the color. Isn't it equivalent to saying that given color, image does not provide any other extra information about the object? -- What is meant by plain answer accuracy in Line 286? -- Why are there missing numbers for few approaches and baselines in tables 2 and 3? -- What is the accuracy metric in table 3? -- Line 340, I don't understand the meaning of this sentence - "the ones that have an "a" in the question ID".

4. Originality The COCO-QA dataset generated automatically from MSCOCO captions dataset is a novel work. The approaches presented in the paper for image question answering are a combination of existing approaches (CNN, LSTM). Such approaches have been explored for the task of image captioning. These approaches are novel in the sense that they answer questions by classifying the answer into a set of predefined classes.

5. Significance The motivation of the paper is sound. The approaches and the baselines presented in the paper throw light on pros and cons of different models for the task of image question answering. If the technical details are provided, this paper can be a good basis for building new models for this task. The COCO-QA dataset introduced in the paper is a novel dataset but it is automatically generated dataset, hence it is limited by the accuracy of the generating algorithm. Also, the questions are derived from the captions dataset, so there will a lot of similar questions since the captions tend to be generic.

Summary: The approaches and baselines presented in the paper for the task of image question answering explore different aspects of the solving the problem. But the paper is missing some important technical details such as what are the answer classes.

The COCO-QA dataset introduced in the paper is a novel dataset but I have concerns about the usefulness of this dataset for the task of image question answering. First, it (potentially) suffers from parsing artifacts. Second, it suffers from the same problem that plague captions -- that captions tend to be generic and only capture the salient objects in the image. Consequently, this dataset does not force a QA system to truly understand an image. This defeats the very purpose of visual/image QA.

Submitted by Assigned_Reviewer_5

Summary.

The paper targets a challenging and recent Image QA problem.

There two main contributions.

First, the authors have proposed a new question answering dataset that is automatically built from the

image captioning dataset COCO.

Second, the authors have proposed a neural network approach to deal with the question answering problem

on two datasets: a reduced Daquar dataset with only single-word answers, and a newly created transformed COCO

dataset. The method significantly outperforms the previous state-of-the-art, and is among one of the

very first neural-based approaches towards Image QA. Moreover, the authors have implemented

a set of baselines

some of them quite interesting like

'deaf', 'blind' or IMG+PRIOR models. Finally the WUPS scores indicate that the proposed

model learnt better

concepts than the other architectures.

Questions.

* Why the authors didn't apply the proposed method on full daquar?

* How difficult it is to use the dataset conversion algorithm with other captioning datasets such

as Flickr30k? How long does it take to convert a dataset? Can the authors provide any

estimates on both questions?

* Can the authors quantify/estimate the quality of the transformed question answer pairs?

* The problem formulation could be more extensive. It is also unclear what are the 'thorny issues that plague

multi-word generation problem'. Can the authors elaborate on it?

* Do you consider questions about subject or object? How do you decide on the type of such question?

How do you convert sentences with many adjectives like

'An elegant pinky man rides a fast and white horse'? What's about sentences with

(non-)restrictive clauses like 'An elegant, weared in a pinky suit, man rides a white horse that is fast'?

* How numbers are extracted from the sentences? Can the procedure extract number of people

in the following sentence: "John and Anne are walking down the stree"? Many such details are

missing from the paper's description.

* Can you provide more statistics on the generated questions like

average length, the most/least common answer (counts and answer word),

the median answer (counts and answer word)? How many answers appears only once in training data?

* I wonder if there are questions in test set that are very similiar to questions in training set, but

have different answer. How much training/test questions overlap?

How much training/test entire question-answer pairs overlap?

* How IMG+BOW combines image features with bag-of-word vectors?

* IMG+PRIOR model requires to identify 'c' and 'o' in Eq 1. How is it done?

* Lines 333-338 provides an interesting analysis about the performance of 'deaf'/'blind' models

on Daquar and COCO-QA. What is the performance of 'deaf' model on Daquar?

Additional Comments.

* The statemet in lines 021-023 is not true as the paper compares the presented method against two other

image QA models.

* 4.2.2.1 repeats 4.2.1 -- those parts require some rewriting

* The paper lacks a broader vision. The introduction and problem formulation are too short.

* It is difficult to quantify the claim in the line 328 based on the model performance.

It may be the case that questions that require the order are at the same time more difficult to answer,

and the model doesn't do good job in combining the image with language.

* [15] and

[31] are tested and trained on multiple-word answers, while this submission only evaluates on a subset (about 99%) which consists of single word answer. Therefore, the results in Table 2 are not directly comparable - but should be rather close as only 1% of the data is missing. this should be clarified or taken into account in the evaluation (e.g. getting penalised for not producing multiple answer words)

Typos:

line 38: interation -> interaction

line 124: formuate -> formulate

Summary: The submission presents a good evaluation and analysis on a fairly new and relatively poor understood task of Image QA . Several baselines are investigated and new method outperforms previous method by a large margin.

However, the paper has some issues, most importantly there is no evaluation of the COCO-QA quality that

should be expected from the automatic conversion methods. The preliminary rating is accept but with reservations subject to the questions posed below.

Author Feedback
Author rebuttal: We thank the reviewers very much for their valuable comments and suggestions. Here, we respond to some of the primary criticisms and questions.

R1, R3 and R6 asked about the generality of our question generator. We would like to point out that our dataset generation algorithm is portable and extensible. It can be directly used on the Flickr 30k dataset with only some minor modifications on exception rules. R3 asked whether it will fail on complex sentence structures, such as "An elegant pinky man rides a fast and white horse." and "An elegant, weared in a pinky suit, man rides a white horse that is fast." The parser result of the 1st sentence is [NP [NP [DT An] [JJ elegant] [NN pinky]] [NP [NN man] [NNS rides]] [NP [DT a] [ADJP [JJ fast] [CC and] [JJ white]] [NN horse]]]]. Due to the parser mistake of identifying "pinky" as a noun, the algorithm did not produce the best question, but still a grammatical one: "An elegant pinky man rides what?" "-Horse." For the 2nd sentence, our software outputs two questions: "What does an elegant, weared in a pinky suit, man rid?" "-Horse", and "What is the color of the horse" "-White." (note that the spelling error of "ride" into "rid" is caused by the WordNet lemmatizer). In general it works on some fairly complex sentences.

R3 asked for the statistics of COCO-QA: max question length is 55, average is 9.65. The most common answers are "two" (2094, 2.65%), "white" (1929, 2.45%), and "red" (1632, 2.07%). The least common are "lake" (12, 0.0152%) "cups" (12, 0.0152%), and "pans" (13, 0.0165%). The median answer is bed (589, 0.748%). Across the entire test set (38948 QAs), 9072 overlap in training questions (23.29%), and 7284 (18.70%) overlap in training question-answer pairs.

R5 noted that the IMG+PRIOR method makes a strong assumption that object and image are conditionally independent. We are not suggesting that this is true. But this is a good baseline, showing whether the model knows that the color is associated with the object in the image or is only from some prior knowledge.

We chose to run experiments on our COCO-QA dataset as well as DAQUAR-37. In full DAQUAR, around 90% of the questions contain one-word answers, so the results will not be directly comparable. Moreover, the blind models' results suggest that DAQUAR-37 is already too difficult for the current set of models.

R5 had some questions about our evaluation terms. "Plain accuracy" means that if the answer exactly matches the ground truth, then a score of 1.0 is assigned, and 0.0 otherwise. The accuracy metric in Table 3 is plain accuracy. Also, there are a few cells left blank in our tables: First, we were not able to run the IMG baseline on DAQUAR because we don't know the question type. Second, we cannot run MULTI-WORLD on COCO-QA because MS-COCO images don't have depth estimates. Third, the details of the model of ASK-NEURON (a parallel work) has not been released yet. We also evaluated our models with some extra questions. In the figures, if a question ID is purely numeric, that means the question is from the original dataset. If it looks like "1111a" then it is a question that we added to the dataset to test robustness.

R2 asked about significance testing of the reported results. On DAQUAR, experiments on the same model have 2-3% of standard deviation. On COCO-QA, results are much less noisy, with ~1% of standard deviation.

R6 suggested utilizing attention in the model, and a nearest-neighbor baseline. We have done some experimentation with models that incorporate attention, but have yet to find any performance gains. The nearest-neighbor suggestion is a good one that we will include.

R2,4,6 commented on the interesting result that the IMG+BOW model has similar or better performance compared to the VIS+LSTM model, and asked for more details on this comparison. The final paper will contain a more thorough analysis, but here are some highlights. First, we tried to give IMG+BOW a 500 dim. image vector, and it does worse than VIS+LSTM (~48%). Since the goal is to report the best performance for each architecture, this result is omitted in the table. However, this provides some evidence that dimensionality reduction does hurt performance. Second, VIS+LSTM with 4096 dim. image vector will mean a 4096 dim. word embedding, which makes the computation much larger, and we did not run experiment on that size. Lastly, by comparing the "blind" version of the BOW and LSTM models, we can test our second observation that word ordering is not critical in the Image QA task. So to summarize, LSTM alone does not yield better language understanding compared to BOW; however, VIS+LSTM is a better combination of the dimension-reduced image features and word vectors, whereas IMG+BOW still works well with the original image features.

Thanks again for the your time reviewing our paper and we will integrate more details in the paper to answer other questions in the reviews.